# Investigation on the Curing and Thermal Properties of Epoxy/Amine/Phthalonitrile Blend

**DOI:** 10.3390/ma17174411

**Published:** 2024-09-07

**Authors:** Cong Peng, Tao Luo, Zhanjun Wu, Shichao Li

**Affiliations:** 1College of Fiber Engineering and Equipment Technology, Jiangnan University, Wuxi 214401, China; 6223017070@stu.jiangnan.edu.cn (T.L.); 8202206002@jiangnan.edu.cn (S.L.); 2College of Textile Science and Engineering, Jiangnan University, Wuxi 214122, China; 3School of Materials Science and Engineering, Dalian University of Technology, Dalian 116024, China

**Keywords:** epoxy resin, phthalonitrile, thermal stability, curing kinetics, pyrolysis

## Abstract

The bisphenol A-type phthalonitrile (BAPH) was blended with the classic epoxy system E51/DDS to prepare the epoxy/phthalonitrile thermoset. The curing kinetics were investigated by differential scanning calorimetry (DSC) using the isoconversional principle, and the average activation energy (E_α_) of the E51/DDS curing reaction was found to decrease from 87 kJ/mol to 68.6 kJ/mol. Combining the results of the rheological study, the promoting effect of phthalonitrile on the crosslink of epoxy/amine is confirmed. The curing reaction of the blended resin was characterized using FTIR, and the results showed that BAPH could react with DDS. The thermal behaviors of the thermosets were investigated via DMA and TGA. The glass transition temperature (T_g_) is found to increase from 181 °C to 195 °C. The char yield increases from 16% to 59.6% at 800 °C in a N_2_ atmosphere, which is higher than the calculated value based on the proportional principle. The AFM phase images show that there is no phase separation in the cured thermoset. The results imply that the cured epoxy/amine/phthalonitrile blend is probably a kind of copolymer. The real-time TG-MS indicated that the pyrolysis of the thermoset can be divided into two relatively independent stages, which can be assigned to the cleavage of the E51/DDS network, and the phthalocyanine/triazine/isoindoline, respectively.

## 1. Introduction

Epoxy resin, which belongs to a major class of engineering polymeric materials, is extensively used as a matrix for polymer composites due to its excellent mechanical properties, superior corrosion resistance, low curing shrinkage, and easy processing [1,2,3,4,5]. However, poor flammability and thermal stability limits its use in those areas which demand high thermal stability and good flame-retardant behavior [3,6,7]. Numerous efforts on promoting the thermal stability of epoxy resin have been reported, including the synthesis of a new resin monomer [5,8,9], the modification of the molecular structure of the existing resin [3,7,10,11], and the incorporation of a reinforcement component [6,7]. The blending reinforcement is the most commonly used method in practical production due to its advantages of low cost, convenient operation, and high efficiency [12,13]. The reinforcement components, mixed with epoxy resin, can generally be divided into two forms, including the particulate fillers and a second-phase modifier, i.e., another polymer [3,5,12]. The application of particulate fillers, especially for inorganic fillers, can endow the polymer matrix with impressive promotion of mechanical, thermal, and electrical behaviors [3,14]. However, the favorable effects are conditional on the good dispersion and strong interface interaction of the fillers. Moreover, there are some disadvantages to using these fillers, like the lack of the processability due to the poor flowability, and the deterioration of mechanical behaviors due to the aggregation of the fillers [14,15].

Polymer blending and copolymerization are well-established approaches for tailoring the properties of polymeric materials for particular applications [12,16]. The obtained materials generally have a useful combination of properties derived from the individual components, which can be easily tuned by changing the blend composition [1,17]. Substantial efforts were made in improving the thermal behavior of epoxy resin with a second-phase modifier [13,14,15]. Therefore, the new classes of polymer, which have good thermal behavior and are compatible with epoxy resin, are of great interest [18,19].

Phthalonitrile is an attractive class of thermosetting polymers possessing remarkable thermal stability, with a long-term working temperature of more than 300 °C [20,21,22,23]. During the past decades, many different phthalonitrile monomers have been synthesized, and the properties of the cured resins and their composites have been studied [24,25,26,27]. The cured materials exhibit many impressive properties, including high glass transition temperatures (T_g_ above 450 °C), outstanding thermal and thermo-oxidative stability, and superior fire resistance [23,28,29]. However, there are some drawbacks that limit its applications [30]. First of all, the post-curing temperature of phthalonitrile is usually up to 400 °C, which is much higher than that of traditional thermosets, such as phenol formaldehyde, epoxy, cyanate ester et al., and is unfavorable for the composite manufacture [5,19,27]. Moreover, the high crosslinking density and the rigidity of the crosslinked network result in obvious brittleness of the cured material [30,31,32].

As far as we know, there are few researchers reporting on the investigation of the influence of phthalonitrile on the properties of epoxy thermosets. The above-mentioned issues inspire us to blend epoxy resin and phthalonitrile, and to investigate the curing process and properties of the cured blend. In addition, there are researchers reporting that the curing reaction of phthalonitrile can be promoted by amine substances, which happen to be the most commonly used curing agent for epoxy [33,34,35,36]. This further implies that the epoxy/phthalonitrile blend is a promising method to form materials with remarkable properties and practical value.

Herein, the typical epoxy system bisphenol A epoxy resin (E51)/diaminodiphenyl sulfone (DDS) and bisphenol A-type phthalonitrile (BAPH) were blended to prepare a new thermosetting resin. The curing kinetics and the properties of the generated thermosets were investigated.

## 2. Experiment

### 2.1. Materials

Bisphenol A-type epoxy resin (E51) with an epoxy value of 0.51~0.57 was purchased from Nantong Xingchen Synthetic Material Co., Ltd, Nantong, China. Furthermore, 4, 4′-Diaminodiphenyl sulfone (DDS, purity > 98%), 4-Nitro-o-phthalonitrile (NPh, purity > 98%), bisphenol A (BPA, purity > 98%), N,N-Dimethylformamide (DMF, purity > 99%), and Anhydrous potassium carbonate (K_2_CO_3_, Purity > 99%) were purchased from Shanghai Aladdin Biochemical Technology Co., Ltd, Shanghai, China. All reagents were used as received without further purification.

### 2.2. Preparation of the Bisphenol A-Type Phthalonitrile (BAPH)

The synthesis of BAPH involves two steps, as shown in Figure 1. Typically, 22.9 g of BPA (0.1 mol), 34.7 g of 4-NPh (0.2 mol), 27.6 g of anhydrous K_2_CO_3_ (0.2 mol), and 50 mL of DMF were sequentially added to a 250 mL three-necked flask. The reaction system was heated to 85 °C under a nitrogen atmosphere, with magnetic stirring for 6 h. The substitution reaction occurred between the nitro group of NPh and the phenolic hydroxyl group of BPA, catalyzed by anhydrous K_2_CO_3_. Finally, the resultant solvent containing BAPH was cooled to room temperature, and the anhydrous potassium carbonate was filtered out. The filtrate is then slowly added dropwise to anhydrous ethanol, precipitating yellow powder. The precipitated solid was washed several times and vacuum-dried at 70 °C for 24 h to obtain the BAPH product with a yield of 91.2%.

### 2.3. Preparation of the Thermosets

As the control group, i.e., the pure E51/DDS, the E51 and DDS were mixed and heated to 130 °C under vigorous stirring for 15~20 min until the DDS completely dissolved. The transparent mixture was then vacuum degassed and cured at 160 °C/2 h, 200 °C/4 h, and 220 °C/2 h. The sample is symbolized as ED in the following discussion.

For the mixed thermosets containing E51, DDS, and BAPH, the E51 and the calculated amount of BAPH monomer were mixed, and the mixture was heated to 200 °C until the BAPH completely dissolved. The mixture was cooled to 130 °C and DDS was incorporated under stirring until the DDS dissolved completely. The mixture was vacuum degassed and cured at 160 °C/2 h, 220 °C/2 h, and 270 °C/4 h, respectively. The cured samples were symbolized as EDPHX, in which ‘E’ represents E51, ‘D’ represents DDS, ‘PH’ represents BAPH, and ‘X’ represents the amount of the BAPH monomer. For example, X = 1 indicates that the mass ratio of BAPH/E51 is 1/9, etc. The compositions of the samples are listed in Table 1.

### 2.4. Characterization

The ^1^H NMR and ^13^C NMR spectra of the synthesized product were acquired using a Bruker (Beijing, China) Avance NEO 400 MHz NMR spectrometer, using deuterated dimethylsulfoxide (DMSO-d6) as the solvent. The Fourier Transform Infrared Spectroscopy (FTIR) spectra were recorded on the Thermo Fisher (Shanghai, China) Nicolet iS10 FTIR spectrometer in KBr pellets between 4000 cm^−1^ and 400 cm^−1^. The differential scanning calorimetry (DSC) curves were obtained using a Mettler (Shanghai, China) DSC 3+ differential scanning calorimeter . The peak fitting of the original DSC curve was carried out using the Origin 2021 software, and the Gaussian-Loren Cross mode was selected as the peak type. The thermogravimetric analysis (TGA) curves were obtained using a Mettler (Shanghai, China) TGA2 thermogravimetric analyzer with a flow rate of 50 mL/min, a temperature range of 30 to 800 °C, and a heating rate of 10 K/min. The rheological properties of the mixed system were tested using an Anton Paar Physica (Shanghai, China) MCR301 rotational rheometer. Gel time tests were conducted at 180 °C with a frequency of 1 Hz. The viscosity characteristics of the blended resin were tested under 60 °C to 180 °C, at a heating rate of 5 K/min, with a frequency of 1 Hz. The volatile products from the pyrolysis process of the thermoset were measured online using thermogravimetric Fourier Transform Infrared Spectroscopy mass spectrometry (TG-FTIR-MS) simultaneous measurement. Approximately 10 mg of the sample were heated in the TG from 30 °C to 800 °C at a heating rate of 10 °C/min, under a N_2_ atmosphere. During the measurement, the cell of TG was flushed with 20 mL/min of N_2_ to maintain an inert condition. The gas products released from the TG were swept immediately to a gas cell for analysis by the FTIR and MS analyzer. The transfer lines and gas cell were maintained at 270 °C to avoid the condensation of the produced gasses. Resolution in FTIR was set at 2 cm^−1^, and the spectral region was from 4000 cm^−1^ to 500 cm^−1^. Subsequently, the gaseous products from pyrolysis were analyzed by the MS analyzer, and the changing of reaction temperatures from 200°C to 800 °C was the focus. In the MS characterization, the ion source temperature was maintained at 250 °C, and the scanning range was from 30 *m*/*z* to 200 *m*/*z*. Dynamic mechanical analysis (DMA) was carried out with a Netzsch (Shenzhen, China) DMA 242E instrument under a double cantilever model with a frequency of 1 Hz and a heating rate of 5 K/min. AFM images were obtained using a Bruker (Beijing, China) Dimension Fastscan AFM microscope in tapping mode. The block of cured resin was cut, and the cross-section was polished using 6000-mesh diamond paste. The AFM images were acquired from the central region of the cross-section.

## 3. Results and Discussion

### 3.1. Characterization of the Synthesized BAPH

The synthesized raw BAPH and relevant cured thermosets were characterized by FTIR, as shown in Figure 1. As shown in Figure 1a, the absorption peak at 3330 cm^−1^ that corresponds to the stretching vibration of the OH group vanishes, and the absorption peak at 2230 cm^−1^ that is attributed to the cyan group was observed in the spectrum of raw BAPH, which indicates that the expected BAPH monomer was successfully synthesized [37,38]. The characteristic absorption peak of the cyan group almost disappears, and there are peaks at around 1010 cm^−1^, 1360 cm^−1^, 1520 cm^−1^, and 1720 cm^−1^, which belong to the characteristic peaks of the N-H stretching and bending vibration in the phthalocyanine ring, the C-N stretching vibration in the triazine and isoindole rings, and the C=N stretching vibration in the isoindole ring, respectively, which are observed in the cured BAPH thermoset [20,35,39]. The results in Figure 1a confirm the successful synthesis and self-crosslink of the BAPH.

It was reported that functional groups containing active hydrogen, such as amino groups, phenolic hydroxyl groups, etc., have a promoting effect on the crosslink of phthalonitrile [34,35]. To investigate the interaction between DDS and BAPH, the mixture of BAPH/DDS was also characterized before and after curing. BAPH and DDS were mixed in a mass ratio of 9:1, and the mixture were cured at 160 °C/2 h, 220 °C/2 h, and 270 °C/4 h. As shown in Figure 1b, the FTIR curve of the cured BAPH/DDS thermoset exhibits exactly the same profile as the pure BAPH thermoset. The absorption peaks around 1100 and 1270 cm^−1^ are assigned to the stretching vibration of the S=O group of DDS, which can be observed in the cured thermoset [4,5,40]. It is noteworthy that the sharp absorption peaks at 3200 to 3450 cm^−1^, corresponding to the NH_2_ groups in DDS, become blunt, which implies that DDS was likely to crosslink with the BAPH via the NH_2_ group. The TGA results of the cured neat BAPH and BAPH/DDS thermosets are shown in Appendix A, which indicates that there are certain differences between the two thermosets. It could be inferred that some of the NH_2_ group in DDS may participate in the ring formation reactions of isoindoine, triazine, and phthalocyanine, as shown in Figure 2, and that the reacted DDS may connect the epoxy and phthalonitrile network to form the copolymer.

^1^HNMR and ^13^CNMR spectra were also employed to further confirm the structure of the BAPH monomer. In the ^1^HNMR spectrum, as shown in Figure 2a, in addition to the signals of DMSO and H_2_O, six resonance signals are observed at 1.7 ppm, 7.12 to 7.14 ppm, 7.36 to 7.38 ppm, 7.36 to 7.37 ppm, 7.78 ppm, and 8.09 to 8.10 ppm, with the integration of 1:1:2:1:2:3; these factors are consistent with the molecular formula of BAPH. For the _13_C spectrum of BAPH, all of the corresponding signals are marked, which can also confirm the molecular formula of BAPH.

### 3.2. Rheological Properties of the E51/DDS/BAPH Blend

Rheological properties significantly affect the processing performance of the resin [41,42]. The rheological curves of the epoxy/amine/phthalonitrile blends with various mass fractions of BAPH were measured, as shown in Figure 3. The viscosity-temperature curves of the blends with different BAPH contents were measured with a heating rate of 5 K/min, and within the temperature range of 60 to 180 °C, as shown in Figure 3a. The incorporation of BAPH into the epoxy system significantly increases the viscosity of the blends at low temperature. The viscosities of EDPH1 and EDPH3 at 70 °C are 0.4 Pa·s, 1200 Pa·s, and 5000 Pa·s, respectively, which differ a lot from the increasing of BAPH content. The viscosity of EDPH1 and EDPH2 decreases to 1 Pa·s at 63 °C and 125 °C, respectively. The viscosity of the blends at 180 °C, corresponding to the heating time, are presented in Figure 3b. It is generally known that the viscosity of the resin mixture containing hardener tends to increase at a certain degree of temperature, which is consistent with the results in Figure 3b. Herein, we introduce an index, t_gel_, which is the time when the viscosity reaches 500 Pa·s. This is shown in Figure 3b. It is noteworthy that the order of t_gel_ is EDPH3 (17.5 min) < EDPH2 (43.7 min) < EDPH1 (116.2 min), which indicates that the mixture including more BAPH has higher apparent activity. Further study on the curing mechanism will be discussed in the following Chapter.

### 3.3. Curing Process of the E51/DDS/BAPH Blend

The curing process of the blends was investigated using DSC. For the blends containing epoxy resin, the E51 and BAPH were mixed at 200 °C until the BAPH completely dissolved in E51. As shown in Figure 4, the DSC curve of the pure BAPH monomer shows a single endotherm attributed to the melt process, and there is no curing exotherm observed. This indicates that the curing reaction of the BAPH monomer does not occur spontaneously, even at a fairly high temperature. By contrast, a sharp exotherm attributed to the crosslink of BAPH and DDS is observed for the BAPH/DDS blend at 264 °C, which is much lower than the curing temperature of the pure BAPH monomer, reported in [33,38]. The blends involving E51 resin were also carried out in the DSC study. It is noteworthy that the mixture shows no melting endotherm for both BAPH and BAPH/E51/DDS. This could be explained by the fact that the BAPH and E51 resin are miscible, while the E51 molecules disrupt the crystallinity of the dissolved BAPH monomers, resulting in the vanishing of the endotherm. The exotherm for BAPH/EP at 362 °C can be attributed to the pyrolysis of the uncrosslinked monomer, and no curing reaction is observed. A broad curing exotherm is observed for the EP/BAPH/DDS blend. This exothermic peak was further investigated via peak fitting, as shown in Figure 4b. The DSC peak of the EP/BAPH/DDS blend can be perfectly (R^2^ > 0.99) divided into three individual peaks at 227 °C, 276 °C, and 392 °C, which can be attributed to the crosslink of E51/DDS, BAPH/DDS, and the decomposition of the cured thermoset, respectively. The peak of the BAPH/DDS crosslink exotherm becomes flat and the temperature increases, which can be attributed to the restraining effect of the crosslinked E51/DDS network on the movement of the BAPH molecule.

The curing kinetics of the E51/DDS/BAPH blend were investigated using isoconversional computational methods based on DSC data to investigate the interaction between BAPH and the epoxy/amine component. The kinetic analysis of thermal transformations is based on the assumption that the transformation rate (*dα/dt*) during a reaction is related to two functions, one of which is the temperature (*T*), while the other is the conversion rate *α*
(1)$$\frac{d\alpha }{dt}=K(T)f(\alpha )$$
in which *K*(*T*) is a temperature-dependent reaction rate constant, and *f*(*α*) is a differential conversion function dependent on the curing mechanism. The temperature-dependent function can be easily determined using the Arrhenius equation, as follows:(2)$$K\left(T\right)=Aexp\left(-\frac{E_{\alpha }}{RT}\right)$$
in which *A* is the pre-exponential factor, *E_α_* is the apparent activation energy, *R* is the universal gas constant, and *T* is the absolute temperature.

The frequently applied methods determining E_α_ are the Kissinger, Ozawa, and Friedman methods for non-isothermal reactions. The Starink method shown in Equation (3) is recommended here due to its higher accuracy of E_α_, according to the ICTAC kinetic project [43].
(3)$$ln\frac{\beta_{i}}{T_{\alpha ,i}^{1.92}}=Const-1.0008\left(\frac{E_{\alpha }}{RT_{\alpha }}\right)$$
in which *T_α,i_* is the temperature at which the extent of conversion *α* is reached under the *‘i’*th heating rate, and *β* is the heating rate.

The conversion rate *α* of the curing reaction was measured using DSC, based on the heat released, as follows:(4)$$\alpha_{T}=\frac{{∆H}_{T}}{{∆H}_{0}}=\frac{\int_{t_{0}}^{t_{T}}m\cdot DSCdt}{\int_{t_{0}}^{t_{1}}m\cdot DSCdt}$$
in which *∆H_T_* is the heat released until the specific temperature, *T*, is reached, and *∆H_0_* is the total heat released during the entire curing stage.

The curing process of BAPH with various amount of DDS was studied in order to reveal the influence of amine on the curing of phthalonitrile. The calculations of E_α_ values corresponding to the BAPH crosslink reaction for the BAPH/DDS systems with 5 wt.%, 10 wt.%, and 15 wt.% DDS are shown in Appendix A. The E_α_ values approximately reveal a downtrend, along with the reaction process, and this implies the autocatalytic mechanism of the curing process. The E_α_ values for non-catalytic phthalonitrile were reported to be 120.6 kJ/mol, and even reaching as high as 175.2 kJ/mol [39,44]. The calculated E_α_ averages of BAPH/DDS blends are 82.0 kJ/mol, 92.2 kJ/mol, and 98.2 kJ/mol, respectively, which are much lower than the reported values, which confirms the catalytic effect of DDS on the curing of BAPH. It is noteworthy that the catalytic effect does not improve with the increase in DDS content. This could be explained through the fact that the optimal dosage of DDS is not so high to promote the curing and the excessive amount of DDS will dilute the BAPH, causing a negative effect.

The DSC curves of the curing process of EDPH2 at heating rates of 5 K/min, 10 K/min, 15 K/min, and 20 K/min were shown in Figure 5a–d. The curing process became complex after the BAPH monomer was added into the E51/DDS system. According to the results shown in Figure 4b, the DSC curves were all perfectly fitted into two individual peaks, which belong to the crosslink reaction of E51/DDS and BAPH/DDS, as shown in Figure 5a–d. For determining the reactivity of the reaction, the activation energy was calculated in the range of α = 0.1-0.9. The values of E_α_ with respect to α were obtained, and they are shown in Figure 5h. The average activating energy of the E51/DDS crosslink reaction was 68.6 kJ/mol, which dramatically deceased compared to that of neat E51/DDS (87 kJ/mol), as shown in Appendix A. Similarly, the average E_α_ values of EDPH1 and EDPH3 are 59.4 kJ/mol and 73 kJ/mol, as shown in Appendix A and Appendix A, respectively. Thus, it can be concluded that the E_α_ corresponding to the E51/DDS crosslink increases with increasing BAPH content, which could be attributed to the steric hindrance effect of the EDPH monomer. Furthermore, it is noteworthy that all of the E_α_ values are lower than those of neat E51/DDS (87 kJ/mol). In the rheology study it was found that the gelation time decreased with the increase in BAPH content; this is consistent with the result of the activating energy herein. It can be inferred that BAPH has a promoting effect on the curing of the epoxy/amine system. This can be explained as follows: the curing reaction between epoxy resin and the amine belongs to the nucleophilic addition reaction, during which the lone-pair electrons on the N atom of the amine attack the C-O bond in the epoxy group and initiate the subsequent reaction. The electron-withdrawing property of cyan reduces the electron density of the C-O bond, and this is advantageous for the nucleophilic attack.

### 3.4. Thermal Behaviors of the E51/DDS/BAPH Thermosets

TGA was applied to investigate the thermal stability of blended thermosets. As shown in Figure 6a, the T_5%_ value increases with the increase in BAPH content in both N_2_ and air atmosphere. The T_5%_ value under the N_2_ atmosphere of cured ED is 363 °C, while that for EDS30 is 390.2 °C. This information can easily be found from the TG curves that the char yield of the blended thermoset increases along with the increasing BAPH mass fraction. Furthermore, the DTG curves provide additional information on the thermal decomposition process. As shown in Figure 6b,d, with the increase in the BAPH content, the peak temperature of the DTG curves increases, and the peak intensity declines dramatically. The maximum weight loss rate of the ED is 1.47%/°C (at 389 °C), which decreases by 80% to 0.3%/°C (at 411.2 °C). The DTG curves of the thermosets in the air atmosphere exhibit two individual peaks which are attributed to the pyrolysis and oxidation processes. The trend of the first peaks is exactly the same as that in Figure 6b, while the situation of the second peaks is different. While the maximum oxidation temperature increases with the increasing BAPH content, the peak weight loss rate at the oxidation stage also increases. This could be attributed to the higher carbon content of the BAPH.

The char yield of the mixture of two individual components can be theoretically calculated via the proportional principle shown in Equation (5), which assumes that the two components decompose independently without interaction.
(5)$$C=(C_{1}\times W_{1}+C_{2}\times W_{2})\times 100\%$$
where *C* is the char yield of the mixed thermoset, *C*_1_, *C*_2_, *W*_1_, and *W*_2_ are the char yields and mass fractions of component 1 and component 2, respectively. Based on Equation (5), the theoretical char yields of EDPH1, EDPH2, and EDPH3 are 21%, 26.2%, and 31.7%, respectively, which are obviously lower than those of the measured data, as listed in Table 2 (the TGA results of BAPH are shown in Appendix A). This result implies that there is an interaction between the E51/DDS and the polyphthalonitrile network, which is consistent with the result discussed in Section 3.1. The results of TGA indicate that the introduction of BAPH into the epoxy/amine system has an obvious effect on promoting the thermal stability of the thermoset.

The pyrolysis test of the thermosets was applied in a tube furnace to provide further information on the thermal stability of the E51/DDS/BAPH thermosets. The cured specimens were heated to 800° C under the N_2_ atmosphere at a heating rate of 10 °C/min in a tube furnace to acquire the residues, as shown in Figure 7. There is an obvious difference between the char appearance of pure E51/DDS and the blended thermosets. The char residue of cured E51/DDS presents an obvious swelling feature, which indicates that the crosslinked molecular skeleton collapsed, and the material melted during heating. By contrast, the cured resin of EDPH1 looks compact, with a barely inflated appearance, and the cured resin of EDPH2 and EDPH3 sustain their exact original appearance. The char images shown in Figure 7b directly demonstrate the excellent thermal stability of curing containing BAPH, which is consistent with the results of TGA.

The volatile products from the pyrolysis process of the cured EDPH3 were measured online by TG-FTIR-MS, as shown in Figure 8. The FTIR results of volatile products in the N_2_ atmosphere during the heating process are shown in Figure 8a. At 300 °C, there is no obvious signal in the spectrum of EDPH2. There are weak signals of CO_2_ (2340 cm^−1^, 2360 cm^−1^) and aromatic compounds at 1520 cm^−1^, when the temperature reaches 360 °C. The typical FTIR signals of the pyrolysis products of H_2_O at 3736 cm^−1^, CO_2_ at 2298 to 2365 cm^−1^, carbonyl compounds at 1710 cm^−1^, and aromatic compounds at 1520 cm^−1^, respectively, cannot be observed until the temperature reaches 360 °C [45], which is consistent with the TGA results. At 400 °C, some new peaks emerged, including phenol derivatives (3650 cm^−1^), methane (3016 cm^−1^), and aromatic ethers (1176 cm^−1^) for both ED and EDPH2. The intensities of the FTIR spectra at 1520 cm^−1^ and 2926 cm^−1^ are summarized as the measurement of the gaseous methane and aromatic compounds during the pyrolysis process. These two classes of gaseous product are the major pyrolysis product of the E51/DDS-based thermoset. As shown in Figure 8b, both of the two classes of product emerged within the temperature range of 360 °C to 800 °C. It was reported that the FTIR absorbance intensity attributed to methane and aromatic compounds of the E51-based thermoset presented one peak, at about 410 °C and 430 °C [45]. By contrast, there are two peaks at 440 °C and 670 °C in the spectrum of aromatic compounds, and there are multiple and overlapping peaks of methane in the spectrum of methane compounds, which indicates that there are at least two classes of crosslined structure in the thermoset. Moreover, it implies that the decomposition of the epoxy/amine-based thermoset containing phthalonitrile is milder and withstands a wider temperature range.

Figure 9 shows the distributions of all of the fragment ions of EDPH2 by full scanning mode at 390 °C, 440 °C, and 670 °C, which are attributed to the temperature of the peak pyrolysis rate of ED in N_2_ (as shown in Figure 6b), and the temperature of the first peak and second peak in Figure 8b, respectively. As shown in Figure 8a, the MS spectrogram is relatively simple, including the basic peak at *m*/*z* = 40, which is the typical fragment ion peak of the benzene ring. The intensity of the MS signals are rather weak, indicating the slight degree of pyrolysis of EDPH2. The other signals of the main volatile products were identified as follows: benzene (*m*/*z* = 66), C_3_H_8_, and CO_2_ (*m*/*z* = 44).

The crosslinked network of the EDPH2 began to collapse at 440 °C, and thus the MS spectrum presented more signals compared to that at 390 °C. The major signals are as follows: benzene (*m*/*z* = 78, 66, 40), phenol (*m*/*z* = 94), and propane (*m*/*z* = 44) [29]; the attributions of the other signals are shown in Figure 3. It can be concluded from the MS spectrum and the schematic diagram shown in Figure 3 that the pyrolysis of EDPH2 at 440 °C is mainly due to the chain cleavage of the E51/DDS network.

When the temperature reached 670 °C, which corresponds to the second peak in Figure 8b, the MS signals became weaker, and two major changes were observed in the spectrum. The peak at *m*/*z* = 65 and *m*/*z* = 67 became much weaker compared to that at 440 °C, while the peak at *m*/*z* = 66 remained strong, which could be attributed to the formation of the pyrrole icon. A new peak at *m*/*z* = 150 was observed, which could be attributed to the pyrolysis product of the indole molecule. From the information discussed above, it can be concluded that the pyrolysis of the E51/DDS/BAPH thermoset can be divided into two relatively independent stages, which can be assigned to the cleavage of the E51/DDS network and the phthalocyanine/triazine/isoindoline, respectively. The hysteresis of decomposition endows the material with good strength and shape stability at a temperature in which the pure E51/DDS can no longer work.

The DMA results of the thermosets containing a different ratio of BAPH are shown in Figure 10. The E′ at the glass state (i.e., E′ at 50 °C) and rubbery state (i.e., E′ at (T_g_ + 40) °C), and the T_g_ values derived from the peaks of Tan ε curves, are also listed in Table 3. The incorporation of BAPH decreases the storage modulus (E′) at the glass state (i.e., E′ at 50 °C). On the contrary, the glass transition temperature (T_g_, which is derived from the peak of Tan δ) rises with the increase in BAPH content. It can be explained that the crosslinked structure of phthalonitrile dilutes the E51/DDS and decreases the crosslink density of the E51/DDS network, which is the major component of the thermoset, resulting in the decrease of the E′ at low temperature. On the other hand, the crosslinked structure of phthalonitrile is much stiffer than that of E51/DDS in a high temperature. Moreover, it was confirmed that the BAPH would be crosslinked with the DDS monomer, as discussed in the FTIR Chapter. As a result, the chain movement of the crosslinked structure is restrained, and the T_g_ increases. The crosslink density can be evaluated via the formula below:
*C* = *E_r_*′/3*RT*(6)
where *C* is the crosslink density and *E_r_*′ is the *E*′ at the rubbery state.

R is the gas constant (8.314 J/mol·K) and T is the absolute temperature at T_g_ +40 °C. Table 3 shows that all of the calculated values of the C of EDPH1 to EDPH3 are higher than ED; this is consistent with the trend of E′_r_.

### 3.5. The Morphology of the Blending Thermoset

The phase distribution of the ED and EDPH2 were investigated using AFM by the trapping mode. As illustrated in Figure 11c, the phase image of ED presents homogeneous features, excluding the influence of morphological factors. Similarly, no second phase was observed in the phase image shown in Figure 11d, which confirms the good miscibility of BAPH with the E51/DDS system. Moreover, it can be concluded that the obtained E51/DDS/BAPH blending thermoset is, probably, a kind of copolymer in which the polyphthalonitrile interacts with the epoxy/amine network on a molecular scale.

## 4. Conclusions

In the presented work, the bisphenol A-type phthalonitrile (BAPH) was blended with the classic epoxy system E51/DDS to prepare the epoxy/phthalonitrile thermoset. The results of curing kinetics derived from DSC show that the average activation energy (E_α_) of the E51/DDS curing reaction was found to decrease from 87 kJ/mol to 68.6 kJ/mol with the incorporation of BAPH. The promoting effect of phthalonitrile on the crosslink of epoxy/amine was also confirmed, according to the rheological results. The char yield increases from 16% to 59.6% at 800 °C in the N_2_ atmosphere, which is higher than the calculated value based on the proportional principle. There is only one T_g_ peak in the DMA curves, which increases from 181 °C to 195 °C, and there is no phase separation in the AFM phase images, which implies that the obtained thermoset may be a kind of copolymer. The real-time TG-MS indicated that the pyrolysis of the thermoset can be divided into two relatively independent stages, which can be assigned to the cleavage of the E51/DDS network and the phthalocyanine/triazine/isoindoline, respectively. The hysteresis of decomposition endows the material with good strength and shape stability at a high temperature in which the pure E51/DDS can no longer work. The results of this paper provide useful information demonstrating that the blending of epoxy/amine/phthalonitrile could be a promising route for the preparation of thermosets which are promising to be used as high-temperature resistant adhesives, the casing of aircraft and engine, insulation materials for power equipment, the IC encapsulation, etc.

## Data Availability

The original contributions presented in the study are included in the article/Appendix A, further inquiries can be directed to the corresponding author.

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
