# Peer review of "Investigation on the Curing and Thermal Properties of Epoxy/Amine/Phthalonitrile Blend"

_materials, 2024, doi:10.3390/ma17174411_

Round 1

Reviewer 1 Report

Comments and Suggestions for Authors

Interesting paper describing search for modification of epoxy resin to increase its thermal stability and flame-retardant properties. In the Introduction correctly is stressed role of strong interface interaction between epoxy resin and modifying additive. It is shown that added phtalonitrile forms kind of copolymer with epoxy resin. In numerous experimental techniques discussed blends are characterized both on chemical level (NMR, C13NMR; FTIR) and thermomechanical properties are also completely discussed (rheology, DSC, TGA, AFM, char yield, pyrolysis). Good paper, complete research well described.

There are several minor editorial mistakes which should be corrected:

Line 14 – should be …reacts….

Line 52; 131; 212; 218; 222; 282; 283; 334; 342; 343; 360; 361; 368; 382; 396 - there is a space between digit numer and unit 0C. This space should be skipped. Note, that in lines 126; 127; Figure 9 and caption of Figure 9 line 378 there is written correctly (without space)

Line 39 – should be …electrical…

Chapter 2.2 lines 84 – 86 – are really all listed additives placed in 100 ml three-necked flask ? I will use 250 ml flask

Author Response

Thank you very much for your careful and responsible inspection on our manuscript. The modification based on your comments are highlighted in red in the revised manuscript.

Comments 1:

Line 14 – should be …reacts….

Response 1:

It is modified as your suggestion.

Comments 2:

Line 52; 131; 212; 218; 222; 282; 283; 334; 342; 343; 360; 361; 368; 382; 396 - there is a space between digit numer and unit 0C. This space should be skipped. Note, that in lines 126; 127; Figure 9 and caption of Figure 9 line 378 there is written correctly (without space)

Response 2:

It is modified as your suggestion.

Comments 3:

Line 39 – should be …electrical…

Response 3:

It is modified as your suggestion.

Comments 4:

Chapter 2.2 lines 84 – 86 – are really all listed additives placed in 100 ml three-necked flask ? I will use 250 ml flask

Response 4:

It is a clerical mistake. ‘100ml’ is modified to ‘250 ml’.

Reviewer 2 Report

Comments and Suggestions for Authors

In the manuscript, the effect of phthalonitrile on epoxy was discussed. The results were well investigated, but further discussion would improve the manuscript more. My comments are as follows:

The activating energies of E51/DDS for ED and EDPH2 were derived. But how about those of EDPH1 and EDPH3? Just showing the result of EDPH2 to conclude that BAPH decreases the activating energy is misleading because there is a possibility that the energy increases for EDPH1 or EDPH3. Please add a discussion of the ratio of BAPH on the activating energy.

In addition, the change in activation energy of BAPH/DDS in relation to the BAPH ratio would be of interest to readers. Please add a comment on it.

In the AFM observation, how the surface was prepared should be mentioned. Did you cut a flat section inside the bulk to make your observations, or did you just look at the surface of the specimen? This point requires in-depth discussion, as the internal and external morphologies are often different.

Author Response

Thank you very much for your careful and responsible inspection on our manuscript. The modification based on your comments are highlighted in red in the revised manuscript.

Comments 1:

The activating energies of E51/DDS for ED and EDPH2 were derived. But how about those of EDPH1 and EDPH3? Just showing the result of EDPH2 to conclude that BAPH decreases the activating energy is misleading because there is a possibility that the energy increases for EDPH1 or EDPH3. Please add a discussion of the ratio of BAPH on the activating energy.

Response 1:

The data of EDPH1 and EDPH3 are supplied in the supplementary file and the discussion are added in page 10, line 292-297.

Comments 2:

In addition, the change in activation energy of BAPH/DDS in relation to the BAPH ratio would be of interest to readers. Please add a comment on it.

Response 2:

The curing kinetics calulation of BAPH/DDS blends with 10wt.% and 15wt.% DDS were added in Figure S4 and Figure S5 in the supplementary file and the relevant discussion was revised in the main text (page 8, line 262-274)

Comments 3:

In the AFM observation, how the surface was prepared should be mentioned. Did you cut a flat section inside the bulk to make your observations, or did you just look at the surface of the specimen? This point requires in-depth discussion, as the internal and external morphologies are often different.

Response 3:

The block of cured resin was cut and the cross-section was polished using diamond paste of 6000 mesh. The AFM images were acquired from the central region of the cross-section. The above description was added in section 2.4, line 148-150.

Reviewer 3 Report

Comments and Suggestions for Authors

The bisphenol A type phthalonitrile (BAPH) was blended with the classic epoxy system E51/DDS to prepare a new kind of thermoset. The curing kinetics was investigated by differential scanning calorimetry (DSC) . Characterization of the copolymer is presented. The paper could be considered for publication after revision.

- Advantages and disadvantages of the new copolymers should be described clearly in conclusions in comparison with other similar published copolymers.

- Chemical structures of the used monomers and of obtained copolymers should be demonstrated.

- Could the copolymers be characterized by solid state NMR spectroscopy ?

-What part of unreacted materials could be extracted from the cross-linked copolymer ?

- Some practical application of the copolymer should be demonstrated.

Author Response

Thank you very much for your careful and responsible inspection on our manuscript. The modification based on your comments are highlighted in red in the revised manuscript.

Comments 1:

- Advantages and disadvantages of the new copolymers should be described clearly in conclusions in comparison with other similar published copolymers.

Response 1:

The comparison with other similar published copolymers is not necessary due to the following reasons:

First of all, this paper focuses on the influence of phthalonitrile on the curing process and thermal properties of epoxy resin. Copolymers with various amount of PN was prepared and the evolution on thermal and curing properties were studied while the optimal formula and relevant properties are not obtained. It is not proper and serious to make the comparison with other reported materials using the obtained data in this paper. Secondly, the performance data is not comprehensive. Many useful data for practical application, for example the mechanical and aging properties, are not included in this paper. It is hard to conclude the advantages and disadvantages based on the limited data. The last but not the least, the contribution of this paper is to deeply and systematically study the mechanism rather than develop a new copolymer with outstanding behaviors. The study provided inspiration and theoretical support for the subsequent development of materials while no new material was obtained.

Comments 2:

- Chemical structures of the used monomers and of obtained copolymers should be demonstrated.

Response 2:

The chemical structures of the stuff used is added in scheme 1. The description about the obtained copolymers was not exact. The formation of copolymer between epoxy and phthalonitrile is only an inference based on the current limited result. The actual curing process could be very complicated. The description of the inferred structure of the copolymer is added in section 3.1, lines 177-181. Since the exact structure of the obtained thermoset need further confirmation, we think that it is not proper to demonstrate the inferred structure of the obtained thermoset. The original description in some sentences were not proper and now we have modified them. (for example, “confirms“ was modified to “implies” in line 19, “is” was modified to “may be” in line 445)

Comments 3:

- Could the copolymers be characterized by solid state NMR spectroscopy ?

Response 3:

The answer is absolutely yes. The solid NMR spectroscopy has been widely used in the study of polymers. For example, Liu [1]used 13C solid NMR to characterize the revolution of ester group before and after decomposition.Cobos[2] used 29Si NMR to investigate the structure of siloxane linkages and the the formation of the siloxane network was confirmed. Sometimes the solid NMR was necessary while for most of times the characterization of a thermoset can be carried out via other more efficient technique like FTIR, Raman spectra, XRD, etc. Within the scope of this article, solid NMR was not necessary

Comments 4:

-What part of unreacted materials could be extracted from the cross-linked copolymer ?

Response 4:

The purpose of this paper is to demonstrate the influence of phthalonitrile on the curing process and thermal properties of epoxy resin. The actual curing detail is very complicated for this system since there could be many side reactions. We are afraid that the answer of your question is far beyond the scope of this paper and we will try our best to make it out in our further study.

Comments 5:

- Some practical application of the copolymer should be demonstrated.

Response 5:

The epoxy with improved thermal behaviors can be used in areas which demand high thermal stability and good flame-retardant behavior as illustrated in the beginning of introduction section. The practical application of this kind of copolymer can’ t be demonstrated in detail since there is hardly any engineering cases for this type of copolymer. We hope that this material can be applied to practical engineering in the future.

Reference:

[1]T. Liu, X. Guo, W. Liu, C. Hao, L. Wang, W. C. Hiscox, C. Liu, C. Jin, J. Xin, J. Zhang. Selective cleavage of ester linkages of anhydride-cured epoxy using a benign method and reuse of the decomposed polymer in new epoxy preparation [J]. Green Chemistry, 2017, 19(18): 4364-4372.

[2]M. Cobos, E. Pagalday, M. Puyadena, X. Cabido, L. Martin, A. Múgica, L. Irusta, A. González. Waterborne hybrid polyurethane coatings containing Casein as sustainable green flame retardant through different synthesis approaches [J]. Progress in Organic Coatings, 2023, 174(

Round 2

Reviewer 3 Report

Comments and Suggestions for Authors

Some practical application of the copolymers should be demonstrated.

Author Response

Comment 1:Some practical application of the copolymer should be demonstrated.

Reply:The practical application of this copolymer was supplied in the conclusion section in lines 457-459.